# A Prototype Sensor System Using Fabricated Piezoelectric Braided Cord for Work-Environment Measurement during Work from Home

**DOI:** 10.3390/mi12080966

**Published:** 2021-08-15

**Authors:** Yoshiro Tajitsu, Jun Takarada, Kohei Takatani, Riku Nakanishi, Hiroki Yanagimoto, Seita Shiomi, Isamu Nakagawa, Ikuo Kawahara, Takuo Nakiri, Saki Shimda, Yoji Shimura, Takuto Nonomura, Kazunori Kojima, Atsuhisa Ikeguch, Kazuhiro Okayama, Tomohiro Sakai, Yuichi Morioka, Mitsuru Takahashi, Kazuki Sugiyama, Rei Nisho, Koji Takeshita

**Affiliations:** 1Electrical Engineering Department, Graduate School of Science and Engineering, Kansai University, Suita Osaka 564-8680, Japan; shimada-sak@nishikawa1566.com (J.T.); kazunori.kojima@g.softbank.co.jp (K.T.); con@revoneo.com (R.N.); ta@teijin-frontier.com (H.Y.); k99453264@kansa-u.ac.jp (S.S.); k99454567@kansa-u.ac.jp (I.N.); k99484565@kansa-u.ac.jp (I.K.); k98554060@kansa-u.ac.jp (T.N.); 2Nishikawa Co., Ltd., Chuo, Tokyo 103-0006, Japan; k98508970@kansa-u.ac.jp (S.S.); Shimura36@nishikawa1566.com (Y.S.); TTNonomura@nishikawa1566.com (T.N.); kazunori.kojima2@g.softbank.co.jp (K.K.); 3SoftBank Corp., Information Technology Division, Minato, Tokyo 105-7529, Japan; Atsuhisa.Ikeguch8@g.softbank.co.jp (A.I.); Kazuhiro.Okayama234@g.softbank.co.jp (K.O.); Tomohiro.Sakai76@g.softbank.co.jp (T.S.); Yuichi.Morioka55@g.softbank.co.jp (Y.M.); 4Revoneo LLC, Fushimi, Kyoto 600-8086, Japan; contactmT@revoneo.com (M.T.); contactks@revoneo.com (K.S.); 5Teijin Frontier Co., Ltd., Kita, Osaka 530-8605, Japan; nishioR7@teijin-frontier.com (R.N.); takeshita-ko34@teijin-frontier.com (K.T.)

**Keywords:** poly-L-lactic acid, piezoelectricity, braided cord, sensing, PLLA

## Abstract

We proposed a new prototype sensor system to understand the workload of employees during telework. The goal of sensing using such a system is to index the degree of stress experienced by employees during work and recognize how to improve their work environment. Currently, to realize this, image processing technology with a Web camera is generally used for vital sign sensing. However, it creates a sense of discomfort at work because of a strong sense of surveillance. To truly evaluate a working environment, it is necessary that an employee be unaware of the sensor system and for the system to be as unobtrusive as possible. To overcome these practical barriers, we have developed a new removable piezoelectric sensor incorporated in a piezoelectric poly-L-lactic acid (PLLA) braided cord. This cord is soft and flexible, and it does not cause any discomfort when attached to the cushion cover sheet. Thus, it was possible to measure the workload of an employee working from home without the employee being aware of the presence of a sensor. Additionally, we developed a system for storing data in a cloud system. We succeeded in acquiring continuous long-term data on the vital signs of employees during telework using this system. The analysis of the data revealed a strong correlation between behavior and stress.

## 1. Introduction

The corona virus disease (COVID-19) pandemic is causing tragedies worldwide. The pandemic is still far from being controlled, and lifestyles and work have changed drastically, such as the practice of telework. Telework has long been recommended as part of work style reform in Japan, but it has not been easily adopted because it often goes against the work style rooted in Japanese culture [1,2]. However, the Covid-19 pandemic inevitably promoted its introduction in Japan. Therefore, it is necessary to urgently evaluate and verify the workload in telework from the viewpoint of not only work efficiency and creativity, but also maintaining a comfortable working environment and the physical and mental health of workers. To realize this, vital sign sensing using wristband-type devices and image processing technology with a web camera are currently being promoted. However, the former creates a sense of discomfort during work and the latter creates a strong sense of being monitored, and each type of technology has considerable problems when used for evaluation. To truly evaluate the comfort of a working environment, it is necessary that the worker is unaware of the presence of a sensor system. We have proposed and put into practical use wearable sensors that can be integrated into daily life such as stitching piezoelectric sensors to ready-made clothes and the use of piezoelectric-fabric-made clothing [3,4,5]. However, we found that these sensors depreciate existing clothing and it also takes time and effort to remake clothes. Therefore, in this study, we developed a new removable piezoelectric poly-L-lactic acid (PLLA) [6,7,8,9,10,11,12,13] braided cord to overcome the above practical barriers. We attached this cord to a cushion cover sheet and measured the workload of an employee working from home. We then developed a prototype system that stores the data obtained over a long period in a cloud system, and we analyzed the data to measure the degree of stress that occurs when working from home. We here report the results obtained.

## 2. Fabrication of Piezoelectric PLLA Embroidery Blade

Ceramic materials are widely used as typical and practical piezoelectric materials in many devices such as si*x*-axis accelerometers [7,11,12]. They are indispensable in modern society. However, in general, many types of ceramic piezoelectric materials containing lead, are hard and brittle, making it difficult to mold into large sizes. To realize actual wearable sensors, we have developed a new piezoelectric material, which is a piezoelectric braided cord made of green plastic PLLA fiber [6,7,8,9,10,11,12,13,14,15] with its piezoelectric properties maintained (hereafter piezoelectric PLLA braided cord: tensile strength ≥ 0.5 cN/dtex, elongation at break ≥ 1%, and electric field strength ≥ 10 kV/cm) [3,4,5,13,14,15,16,17,18,19]. One of the features of the piezoelectric PLLA braided cord is that it can be sewn directly into the fabric or tied depending on the “softness” and “thinness” of piezoelectric PLLA fibers [13,14,15,16,18], examples of which are shown in Figure 1. Due to the low relative permittivity *ε*′ of PLLA (*ε*′ = 3), the piezoelectric *g* constant is high (*g*_14_ = 0.22 Vm/N). Therefore, this piezoelectric PLLA braided cord is an excellent material as a sensor. Applying a shear stress of 1 N/mm^2^ to the piezoelectric PLLA braided cord yields an output voltage of approximately 0.1 mV. In addition, an output voltage of about 1 mV can be obtained with a bending stress of 1 N/mm^2^. On the other hand, even if a tensile stress is applied to the piezoelectric PLLA braided cord, the piezoelectric response cannot be observed. The piezoelectric PLLA braided cord goes well with soft fibers, such as those in clothes, and can be combined with different materials. Another feature of the piezoelectric PLLA braided cord is that it can selectively detect the directions of stress and strain and enables the selection of the types of stress and strain to be detected by using appropriate knots and embroidery stiches. Therefore, there is no need for a circuit system or a personal computer (PC) that requires complex calculations for processing because sensing is the only required procedure. Taking these features together, piezoelectric PLLA braided cords can be used as sensors that can detect expansion, contraction, shear, and bending, depending on how they are tied or embroidered with the type of stich pattern [3,4,5,20,21,22].

### 2.1. Piezoelectric PLLA Braided Cord

We considered attaching a piezoelectric PLLA braided cord as a sensor to a cushion placed on a chair used at home when working from home. The idea was to embroider cushion covers with the piezoelectric PLLA braided cord, which had been studied already for a long time (an example of embroidery with the PLLA braided cord is shown in Figure 2 and used as a sensor) [3,4,5,20,21] However, with this ideas, in order to embroider the sheets, it is necessary to collect them from each home and then prepare them according to the shape of each chair before experiments on vital sign sensing during telework can be conducted. This takes too much time and effort to prepare, and clinical trials cannot be conducted. Therefore, we attempted to develop a piezoelectric PLLA braided cord sensor on a tape that can be attached to each cushion cover as it is. The knowledge gained from our studies on piezoelectric PLLA embroidery is useful for the design of the motion sensor used here [3,4,5,20,21]. When the piezoelectric PLLA braided cord is attached to clothing and used as a sensor, the sensor does not directly bear the weight; however, in this setup, a static load such as human weight is applied to the cushion of the chair. That is, the load is always applied vertically to the cord sensor. This makes a large difference in the design of the sensor [3,4,5,20,21]. For example, if the piezoelectric PLLA braided cord is under the weight of the person sitting on the chair, the braided cord may be displaced from its initial position on the chair for measurements, which should cause abnormal conditions that should be prevented [3,4,5,20,21]. The basic condition that ensures the stability of sensing is to provide a safe, easy, and reliable means of mounting so that anyone can use it easily. Therefore, we decided to design a tape-shaped piezoelectric PLLA braided cord sensor that can be easily attached to and detached from a sheet.

### 2.2. Finite Element Method (FEM) Calculation

The piezoelectric PLLA braided cord exhibits a large piezoelectric response to bending displacement [3,4,5,20,21,22]. Therefore, depending on the spacing and placement of bending points on the braided cord and the symmetry of the placement, the piezoelectric response is only detectable in a specific direction of the applied stress [3,4,5,20,21,22]. For the piezoelectric PLLA embroidery designed on this basis, we previously reported that the chain stitch structure efficiently produces piezoelectric responses to various stimuli such as tension, bending, and compression [3,4,5,20,21,22]. In an embroidery pattern, in general, a piezoelectric PLLA braided cord has a part with multiple points that bend or curve steeply and a connection part between two bending points, and the cord is arranged so as to fill spaces between adjacent segments. A thread is temporary stitched on a fabric to form the desired embroidery pattern. At a point of the fabric with the embroidered pattern, the generated piezoelectric response is large. However, when used on a chair, this point is pressed against the chair by the weight of the person and is constrained between the upper surface of the chair and the thighs. A load of 70 kgW was applied to that point, and the piezoelectric response was calculated by FEM. We used the simulation software which is available on the market (Femtet: Murata Software, Kyoto, Japan), based on the FEM. The simulation software solves some physical phenomena (electric field, mechanical stress, and piezoelectricity etc.). The piezoelectric property values were used for calculating with FEM, as shown in Equations (1)–(3). As shown in Figure 3, the types of stress are tension, shear, and torsion. The figures shows that the piezoelectric response at the point was 1/100 or less, and no signal generation could be expected. Since the design is based on placing the piezoelectric braid along the plane, we considered the type of placement. However, even in such a pattern described above, when static pressure (i.e., body weight) is applied, almost no piezoelectric signal is generated even when the fabric is deformed. Therefore, we observed the magnitude of the response signal of the piezoelectric braid with slightly different heights at both ends in the radial direction. When the step was 3% or more of the radius of the piezoelectric braid, the amplitude magnitude of the response signal could be observed, as shown in Figure 4. From this result, we designed a flat oval PLLA embroidery that could be put to practical use on cushion covers. In the design, we expected that steps would naturally occur at both ends of a piezoelectric braided cord with slightly different heights realized by adjusting the diameter of the embroidery thread around the piezoelectric braid. On the basis of the structure that supports both ends, which could be analyzed by FEM, many experiments by trial and error were carried out to arrive at the design shown in the upper panel of Figure 5. FEM was carried out to confirm the piezoelectric response under static load. As a result, the piezoelectric PLLA braided cord showed a piezoelectric response and can therefore function as a sensor.

Equation (1): Relative dielectric tensor *ε_ij_*
(1)$$\varepsilon_{ij}=\left(\begin{array}{ccc} 2.7 & 0.0 & 0.0\\ 0.0 & 2.7 & 0.0\\ 0.0 & 0.0 & 2.7 \end{array}\right)$$

Equation (2): Piezoelectric tensor *d_ij_* (pC/N)
(2)$$d_{ij}=\left(\begin{array}{cccccc} 0.0 & 0.0 & 0.0 & 6.3 & 0.0 & 0.0\\ 0.0 & 0.0 & 0.0 & -6.3 & 0.0 & 0.0\\ 0.0 & 0.0 & 0.0 & 0.0 & 0.0 & 0.0 \end{array}\right)$$

Equation (3): Elastic compliance tensor *s_ij_* (10^−10^Pa^−1^)
(3)$$s_{ij}=\left(\begin{array}{cccccc} 3.33 & 1.66 & 0.85 & 0.00 & 0.00 & 0.00\\ 1.66 & 3.33 & 0.85 & 0.00 & 0.00 & 0.00\\ 0.85 & 0.85 & 1.69 & 0.00 & 0.00 & 0.00\\ 0.00 & 0.00 & 0.00 & 7.78 & 0.00 & 0.00\\ 0.00 & 0.00 & 0.00 & 0.00 & 7.78 & 0.00\\ 0.00 & 0.00 & 0.00 & 0.00 & 0.00 & 7.78 \end{array}\right)$$

### 2.3. Trial Production of Piezoelectric PLLA Embroidery Blade

We have searched for a way to achieve the appropriate embroidery pattern. As a result, we found that embroidery blade, such as embroidery lace and knitting of embroidery braided cords, could achieve this [23,24]. Embroidery blade has a long history, and bobbins were used and to produce these blades in ancient Egypt and ancient Rome. In modern times, patterns are created by entwining, assembling, twisting, and tying dozens to hundreds of threads wound around the fulcrum in a bobbin case. It is usually made continuously with the same number of threads from the beginning to the end. Using this technique, one can produce the piezoelectric PLLA embroidery blade structure designed by FEM using piezoelectric braided cords, as one of the strings in a bobbin. The lower panels in Figure 5 show two kinds of piezoelectric PLLA embroidery blade. The left panel shows one of the tape types and the right panel of the string type.

## 3. Experiments

### 3.1. Preliminary Experiment

Before conducting a long-term test, we conducted a preliminary experiment to verify the appropriateness of the design of the developed piezoelectric PLLA embroidery blade. If the subject sits on the cord and feels uncomfortable, it may give the subject additional stress when working, which defeats the purpose of the stress sensor. Therefore, although in Figure 6 the cord is designed to be exposed to show the position, it was actually put inside the seat cover and devised so as not to reduce the original sitting comfort. It was also attached to the backrest seat, if there is one, as shown in Figure 6. Figure 7 shows the analog circuit system used for the piezoelectric PLLA embroidery blade. Next, to confirm the detection accuracy of this piezoelectric PLLA embroidery blade, a male subject was instructed to sit on the chair with this sensor system to measure his vital signs. The subject felt comfortable sitting on the chair so much so that he did not notice the presence of the sensor. Furthermore, a medical electrocardiography (ECG) device was attached to the subject. The simultaneous measurements using the ECG device and piezoelectric PLLA embroidery blade performed in the resting state. During test measurements, the subject did not make large movements to stand up. Typical waveforms are shown in Figure 8. The lower waveform measured by the ECG device shows sharp R wave peaks corresponding to heartbeats, and the upper waveform of response signals from the piezoelectric PLLA embroidery blade also shows R wave peaks. We can see that the waveforms match well. Figure 9 shows the waveforms obtained simultaneously using a respiratory rate meter and the piezoelectric PLLA embroidery blade. Although the waveforms differed, large peaks corresponding to peak intervals were observed, indicating a good match. These results also show that this system can clearly show the workload during telework without causing any discomfort to the subject. On the basis of these results, we proceeded to the main experiment.

### 3.2. Experiment in Environment Close to Actual One

In order to apply it to improve the working environment, it is necessary to continuously accumulate data for at least 8 h/day and 40 h/week for three months. In that case, 100 GB data per week must be processed for each subject, which may be impossible to analyze automatically using a stand-alone PC system when the number of subjects becomes more than ten. Therefore, we developed a prototype system for storing and analyzing data in a cloud system as shown in Figure 10. Data were acquired by the sensor every 10 ms and sent to the gateway by a Bluetooth low energy (BLE) device every 100 ms and to the data server via a net line.

## 4. Results and Discussion

### 4.1. Experiment in Environment Close to Actual One

With the above system we developed, we can share data nationwide and overseas as long as an internet connection is available. Figure 11 shows an example of stored vital sign data at the Kansai University server in Osaka obtained from the vital sign sensing test conducted in Tokyo, which is located more than 400 km west of Osaka. The piezoelectric PLLA embroidery blade can measure not only body movement but also pulse waves, in long-term measurements. On the time scale shown in Figure 11, we observed body movements but we were unable to confirm the changes in pulse waves that occurred in seconds. Thus, we spent much time in analyzing the pulse waves by dividing the data every 20 s. Figure 12 shows the typical data obtained. The pulse rate per minute was converted from the peak pulse rate. As a result, we confirmed on the seconds-time scale that the newly developed sensor system does not require the removal of body movements and myoelectric noises from the original response signal in order to obtain clear pulse waves corresponding to ECG signals. Furthermore, using this system, we obtained signals corresponding to the breathing movement. Figure 13 shows the conversion of these peak intervals into respiratory rates. The signals obtained from the piezoelectric PLLA embroidery blade show that both body movements and pulse waves can be measured. As a result, we could confirm that both parameters can be reliably detected even when the subject’s position changed [25,26,27,28,29,30]. Highly reliable long-term measurement data were obtained using the developed system.

### 4.2. Stress Check

We calculated the stress levels specified in medical books. We analyzed the data obtained using our developed system [25,26,27,28,29,30]. First, the stress levels calculated here are briefly summarized below. The stress level used medically is calculated using the vital sign data obtained during telework [25,26,27,28,29,30]. For stress level, it is first necessary to obtain heart rate data [25,26,27,28,29,30]. The interval time between the peaks of R wave (RRI) is measured, as shown in the top and middle panels of Figure 14 [26,27,28,29,30]. Next, we calculated the power spectrum of these data. The frequency band from 0.04 to 0.15 Hz is defined as LF and the frequency band from 0.15 to 0.40 Hz is defined as HF, as shown in the bottom panel of Figure 14 [25,26,27,28,29,30]. LF levels have been shown to reflect both (vasomotor) sympathetic and parasympathetic nerve activities [26,27,28,29,30]. LF is also known as the “respiratory zone” because it indicates the variation in RRI due to respiration. HF reflects the activity of the parasympathetic nerve (vagus nerve) [26,27,28,29,30]. The ratio of LF to HF (LF/HF) represents the overall balance of sympathetic and pathetic nerve activities. A high ratio indicates sympathetic nerve activity dominance and a low ratio indicates pathetic nerve activity dominance; the higher the ratio, the higher the stress level [25,26,27,28,29,30].

We show a typical example of LF/HF during telework in Figure 15. During measurements, the subjects were preparing materials for the conference, making presentations during the web conference, and having meetings at home. The LF/HF did not change significantly during the preparation of materials. However, at web conferences, the LF/HF increased at the beginning of the presentation. The stress level became stable as the presentation progressed. The stress level increased during discussions. Another, an example is shown in which a female elementary school student in a lower grade did her homework and then played games using her smartphone, as shown in Figure 16. Her LF/HF slightly increased before doing her homework but became almost stable as she was doing her homework. Her stress level (LF/HF) became high occasionally during playing a game. Interestingly, her stress level (LF/HF) markedly increased when her mother told her to quit the game. Currently, this method is unreliable as there are only two measurement examples. In the future, we plan to collaborate with an industrial physician to verify the accuracy of this measurement by comparing it with the physiological hormone measurement sensor value that indicates stress.

The above results indicate that there seems to be a strong correlation between their actual behaviors and stress levels calculated using our system. The results suggest that our sensor system with the piezoelectric PLLA embroidery blade we developed is a very promising tool for grasping the true stress level at home and obtaining data useful for improving the working environment. Currently, we are working with an industrial physician to analyze data and behavior, and we are now preparing for clinical trials. In the future, by increasing the number of subjects, we would like to clarify in detail the points to be solved, such as differences in physical conditions, e.g., physical movement and physique, and to make the system really easy to use.

## 5. Conclusions

We have developed a prototype system with a piezoelectric PLLA embroidery blade as a sensor to provide a comfortable working environment while working from home. This allows the subject to measure the degree of stress while working from home for a long time without discomfort. The prototype system has worked well over a long time. However, our development is just beginning. In the future, in order to truly put this device to practical use, it will be necessary to carry out long-duration work under stable to increase the number of subjects and confirm the effectiveness of this device in collaboration with medical experts. We are determined to overcome this difficult process and realize a high-quality device that can improve the telework environment.

## Figures and Tables

**Figure 1 micromachines-12-00966-f001:**
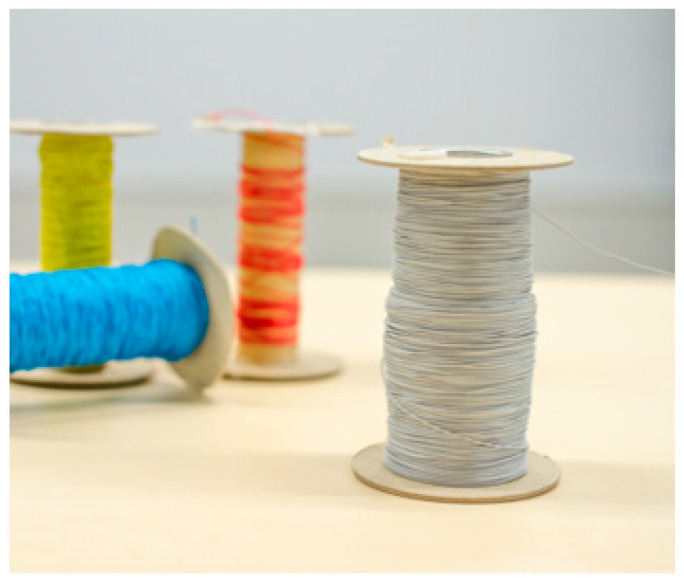
Piezoelectric PLLA braided cord.

**Figure 2 micromachines-12-00966-f002:**
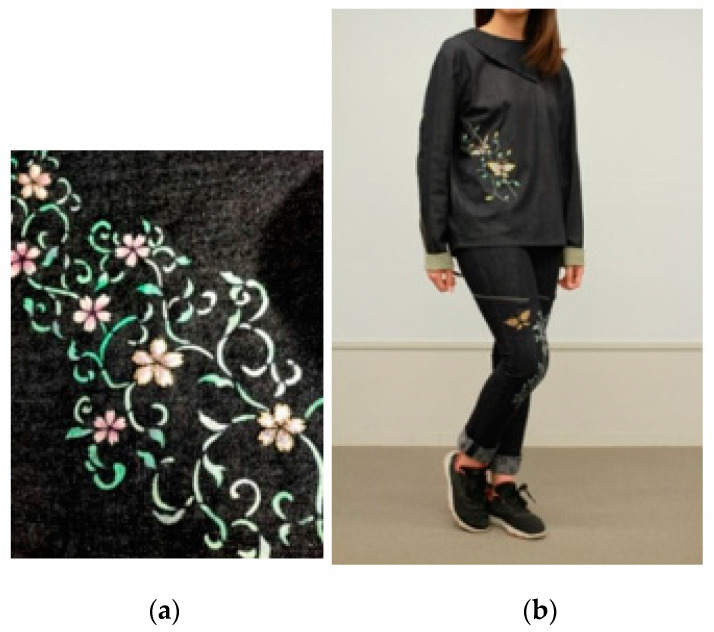
Embroidery with piezoelectric PLLA braided cord: (**a**) Piezoelectric PLLA embroidery pattern for sensing body motion (**b**) Denim pants and jacket with embroidered piezoelectric PLLA braided cord for women.

**Figure 3 micromachines-12-00966-f003:**
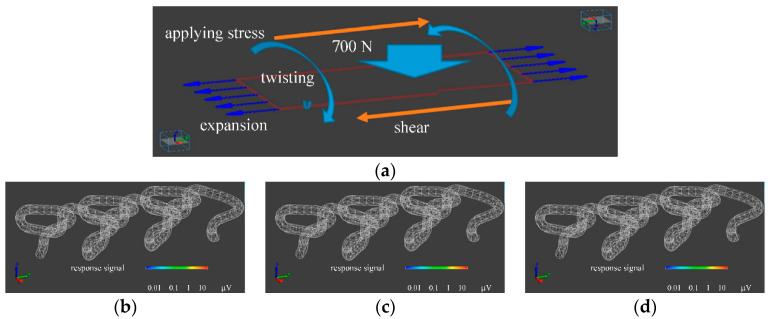
Response signal under the application of stress calculated by FEM. (**a**) Types of applying stress. (**b**) expansion (**c**) shear (**d**) twisting. Piezoelectric response signals could not be obtained in all above cases (No color by RGB pixels is corresponding to generate no piezoelectric response signal).

**Figure 4 micromachines-12-00966-f004:**
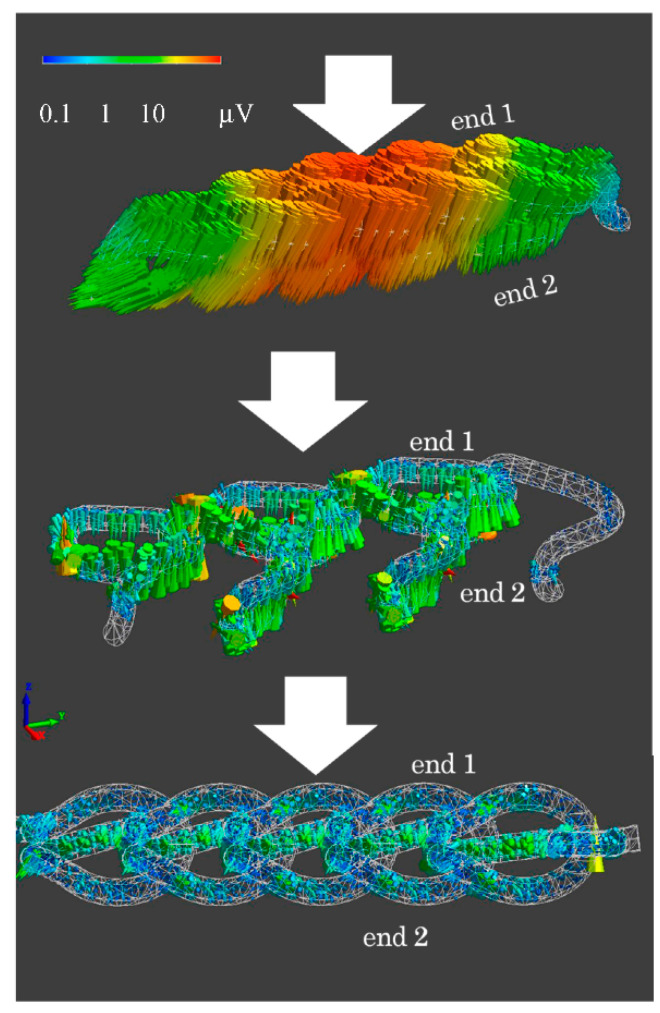
Typical FEM calculation results of response signal in the case of creating height difference between both ends.

**Figure 5 micromachines-12-00966-f005:**
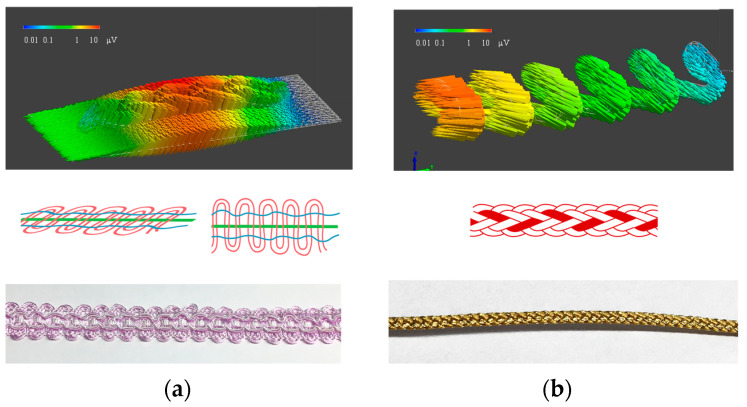
Two types of piezoelectric PLLA embroidery blade developed and their piezoelectric response signals calculated by FEM. (**a**) tape type (**b**) string type.

**Figure 6 micromachines-12-00966-f006:**
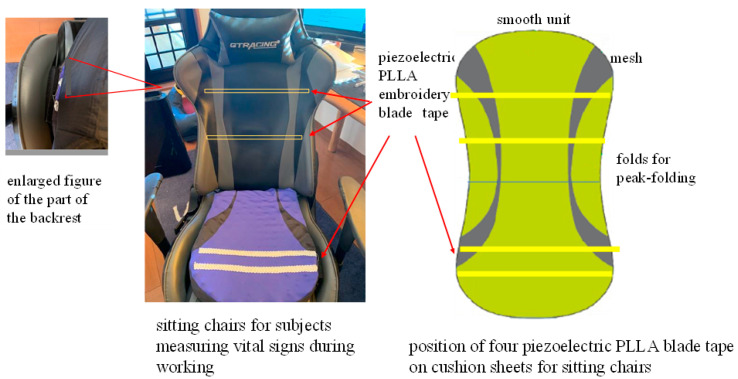
Vital sign sensor system with tape- and string-type piezoelectric PLLA embroidery blades sewn into the cushion sheets of the seat chair.

**Figure 7 micromachines-12-00966-f007:**
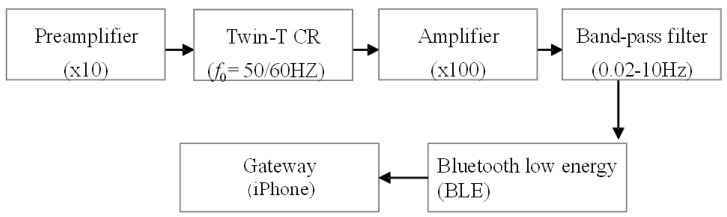
Circuit system for piezoelectric PLLA embroidery blade.

**Figure 8 micromachines-12-00966-f008:**
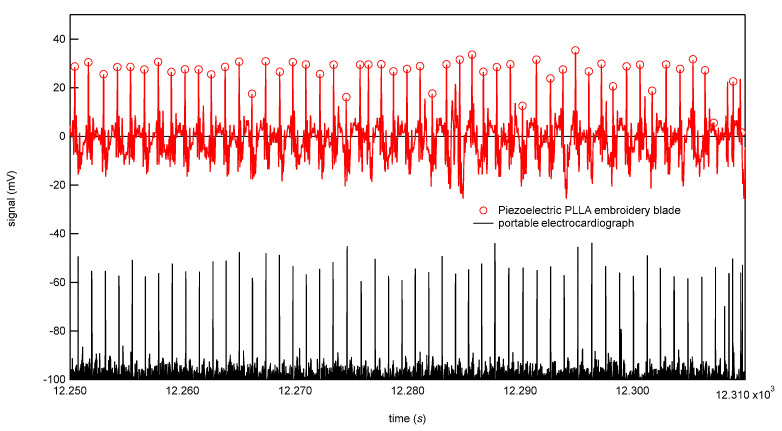
Simaluteous RRI measurement using portable ECG device and piezoelectric PLLA embroidery blade.

**Figure 9 micromachines-12-00966-f009:**
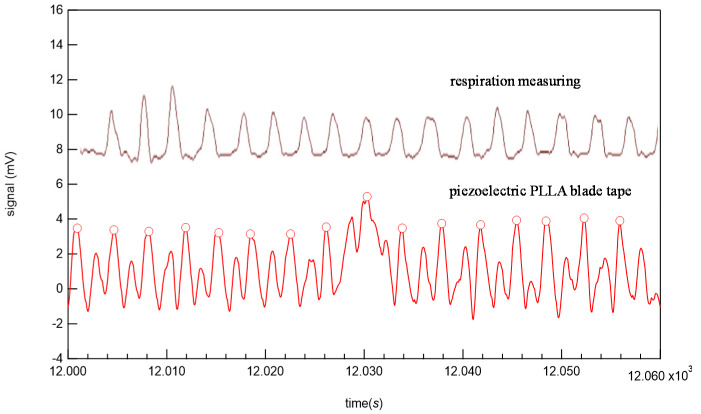
Simultaneously measurements with respiratory rate measurement system and piezoelectric PLLA embroidery blade.

**Figure 10 micromachines-12-00966-f010:**
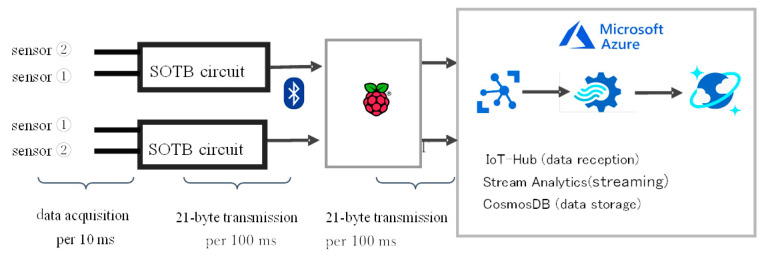
Data storage system for prototype vital sign sensing system for evaluating stress level of worker during telework.

**Figure 11 micromachines-12-00966-f011:**
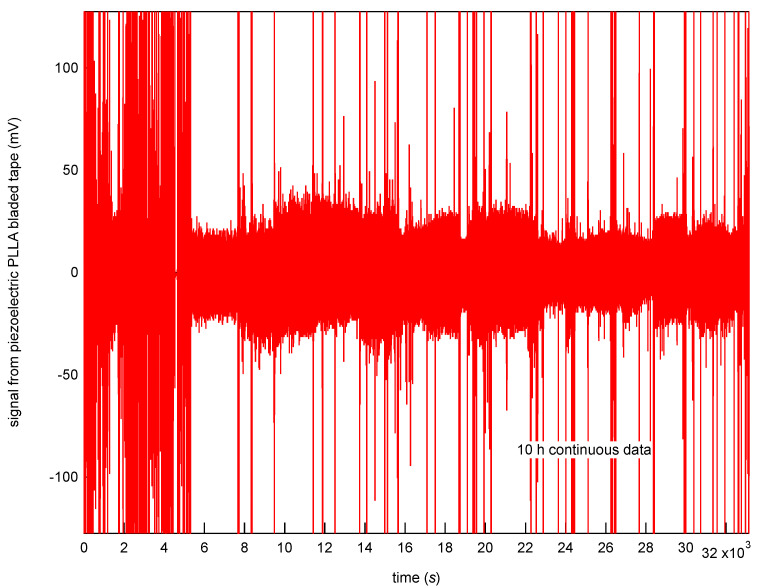
Long-term data obtained using a piezoelectric PLLA embroidery blade sensor system.

**Figure 12 micromachines-12-00966-f012:**
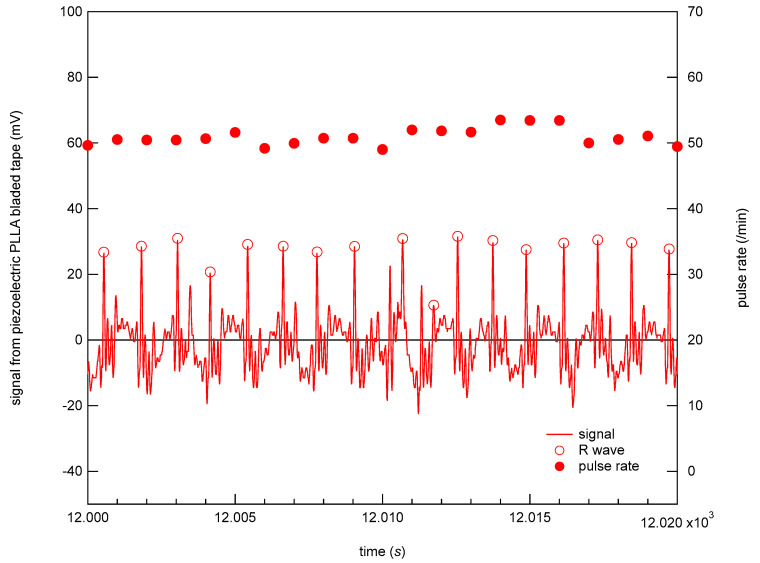
Typical experimental data.

**Figure 13 micromachines-12-00966-f013:**
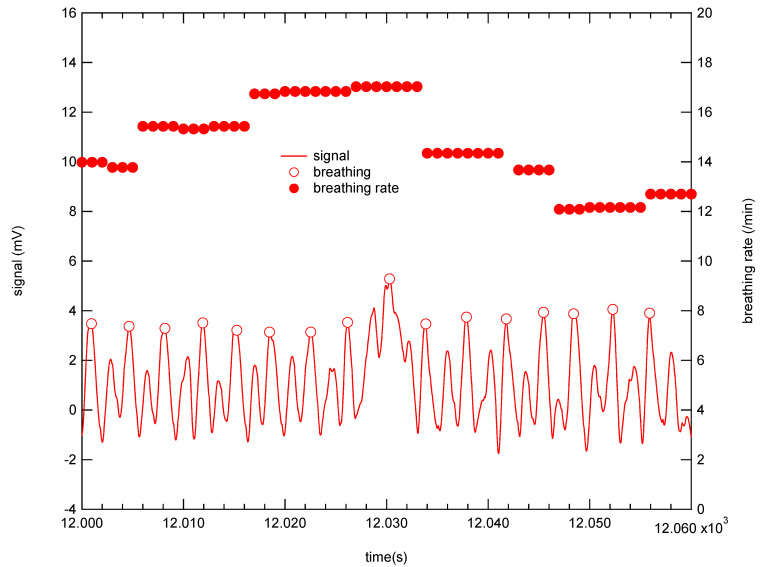
Typical experimental response signals corresponding to respiratory rates from piezoelectric PLLA embroidery blade sewn to cushion cover.

**Figure 14 micromachines-12-00966-f014:**
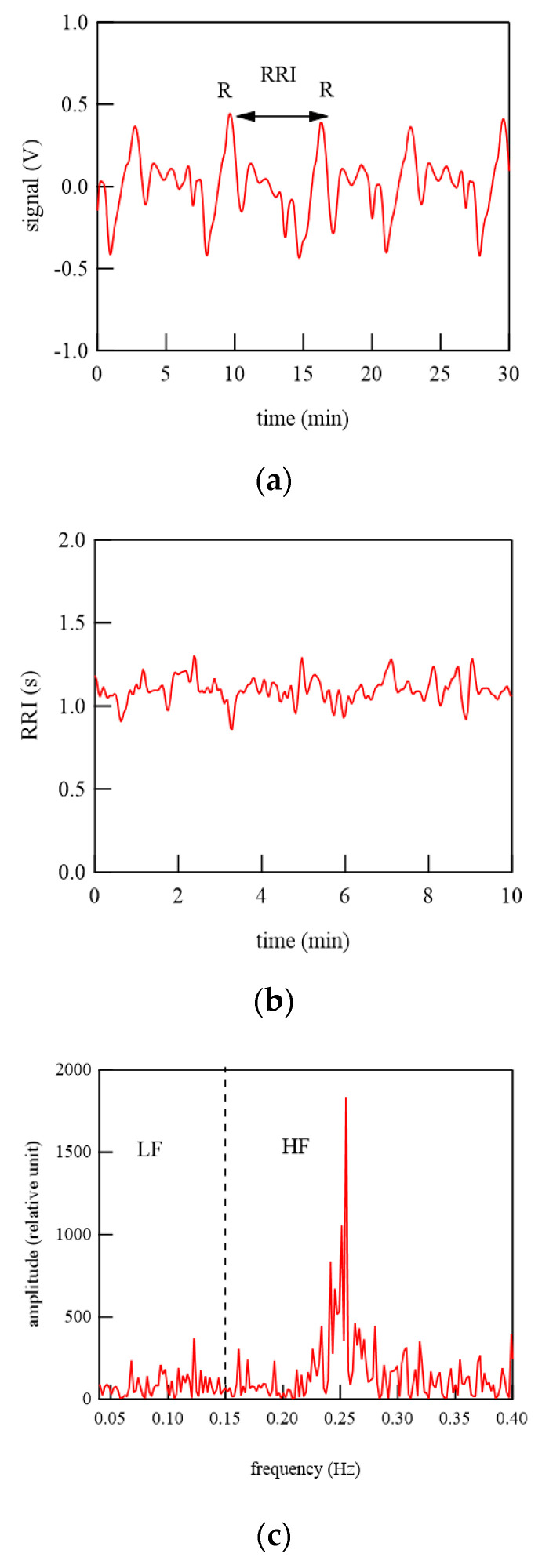
Spectra of RRI. (**a**) pulse wave (**b**) RRI data (**c**) Fourier spectrum of RRI.

**Figure 15 micromachines-12-00966-f015:**
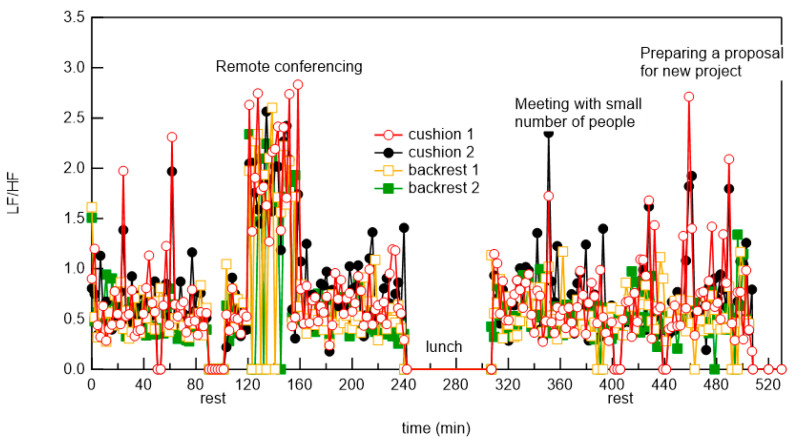
Representative LF/HF measurement results of businessman during telework.

**Figure 16 micromachines-12-00966-f016:**
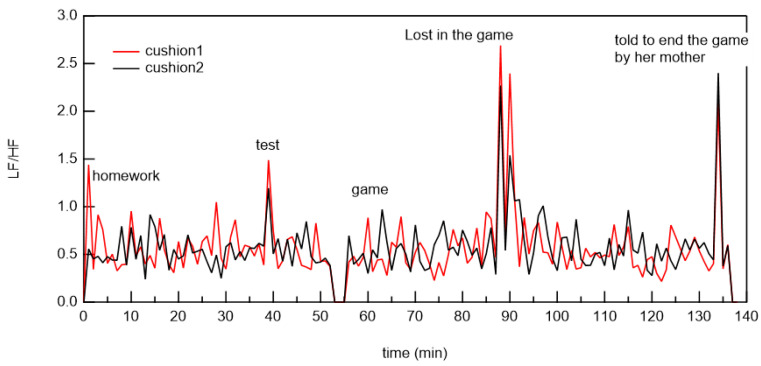
Typical LF/HF of elementary school girl during doing her homework and playing a game on her smart phone after dinner.

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
