# Peer review of "A Prototype Sensor System Using Fabricated Piezoelectric Braided Cord for Work-Environment Measurement during Work from Home"

_micromachines, 2021, doi:10.3390/mi12080966_

Round 1

Reviewer 1 Report

Thank you very much for introducing this good application case of piezoelectric-fiber-based wear sensors. This paper is well orgainized and wriiten. The reviewer recommends the acceptance of this paper, and is still glad if these information can be found in the final version. 

  1. Please introduce some basic information about the piezoelectric fiber used, for example, the sensitivity of this fiber to longitiudinal, shear, torsional, twist deformation. I think these information would be useful for fabricating/inventing more pratical wearable sensors.
  2. Please add some information about FEM simulation. In addition, the color bar in figure 3 may be modified. 

Author Response

We acknowledge your helpful and mindful comments.

1.Please introduce some basic information about the piezoelectric fiber used, for example, the sensitivity of this fiber to longitiudinal, shear, torsional, twist deformation. I think these information would be useful for fabricating/inventing more pratical wearable sensors.

We are revising that in accordance to your guidance.(page 2, lines 66-76)

2.Please add some information about FEM simulation. In addition, the color bar in figure 3 may be modified. 

We corrected the graph. (page 5, Figure 3)

I really appreciate you pointing it out.

Reviewer 2 Report

- On page 3, Figure 2, please write the description titles for the two images as (a) and (b).

- On page 3, Section 2.2, a finite element analysis for piezoelectric PLLA was performed. However, there is no explanation about the physical properties of piezoelectric PLLA, piezoelectric property values, finite element analysis software, etc., so please supplement it.

- On page 4, Fig. 3, the pictorial explanations for loads and generated stresses are difficult to understand. And the color of the charge generation contour is not clearly visible.

- On page 5, Figure 5, please write the title of the figure in (a) tape type and (b) string type. And please explain each image.

- I would like to know whether mechanical and electrical tensile tests were performed on the specimens of piezoelectric PLLA embroidery blades in the experiment on page 6, section 3. Once you have taken the test, please present the results.

- On page 7, there is no physical unit in the title of the vertical axis in Figure 8

- The font format of Figures 7 and 8 is different.

- On page 8, there is no physical unit in the title of the vertical axis in Figure 9

- There are no physical units in the titles of the vertical axes of Figures 11 and 12 on page 9

- On page 11, Figure 14, please write the titles of the three figures as (a), (b), and (c).

- Requires correction on page 10, column 245. the frequency band from above 0.15 Hz is defined as HF

- On page 12, it is interesting to note that the LF/HF values indicate the degree of stress. However, I think this method is unreliable as only two measurement examples. It would be good to compare it with the physiological hormone measurement sensor value indicating stress.

Author Response

We acknowledge your helpful and mindful comments.

  • - On page 3, Figure 2, please write the description titles for the two images as (a) and (b).

We corrected that text. 

  • - On page 3, Section 2.2, a finite element analysis for piezoelectric PLLA was performed. However, there is no explanation about the physical properties of piezoelectric PLLA, piezoelectric property values, finite element analysis software, etc., so please supplement it.

We are revising that in accordance to your guidance.(page 4, line132-136)

  • - On page 4, Fig. 3, the pictorial explanations for loads and generated stresses are difficult to understand. And the color of the charge generation contour is not clearly visible.

We corrected the graph. (page 5, Figure3)

  • - On page 5, Figure 5, please write the title of the figure in (a) tape type and (b) string type. And please explain each image.

We corrected that text. 

  • - I would like to know whether mechanical and electrical tensile tests were performed on the specimens of piezoelectric PLLA embroidery blades in the experiment on page 6, section 3. Once you have taken the test, please present the results.

We add an explanation to the text.(page2 line 66-76)

  • - On page 7, there is no physical unit in the title of the vertical axis in Figure 8

We corrected the graph. 

  • - The font format of Figures 7 and 8 is different.

We corrected the graph. 

  • - On page 8, there is no physical unit in the title of the vertical axis in Figure 9

We corrected the graph. 

  • - There are no physical units in the titles of the vertical axes of Figures 11 and 12 on page 9

We corrected the graph. 

  • - On page 11, Figure 14, please write the titles of the three figures as (a), (b), and (c).

We corrected the graph. 

  • - Requires correction on page 10, column 245. the frequency band from above 0.15 Hz is defined as HF

We corrected that text.(page11 line 276)

  • - On page 12, it is interesting to note that the LF/HF values indicate the degree of stress. However, I think this method is unreliable as only two measurement examples. It would be good to compare it with the physiological hormone measurement sensor value indicating stress

We are revising that in accordance to your guidance.(page12 line295-300)

We really appreciate you pointing it out.